# Optimization of Processing Parameters for Water-Jet-Assisted Laser Etching of Polycrystalline Silicon

**Xuehui Chen, Xiang Li, Chao Wu, Yuping Ma, Yao Zhang, Lei Huang * and Wei Liu**

College of Mechanical and Electrical Engineering, Anhui Jianzhu University, Hefei 230601, China; xhenxh@163.com (X.C.); lix1117@163.com (X.L.); 15212428852@163.com (C.W.); wxlmyp@163.com (Y.M.); nianmin134@163.com (Y.Z.); wliu@hfcas.ac.cn (W.L.)

* Correspondence: huangl75@ahjzu.edu.cn; Tel.: +86-137-2110-2968

**Abstract:** Liquid-assisted laser technology is used to etch defect-free materials for high-precision electronics and machinery. This study investigates water-jet-assisted laser etching of polysilicon material. The depths and widths of the etched grooves were investigated for different water-jet incident angles and velocities. To select optimal parameters for a composite etching processing, the results of many tests must be compared, and at least one set of good processing parameter combinations must be identified. Herein, the influence of different parameters on the processing results is studied using an orthogonal test method. The results demonstrate that the depths and widths of the processing grooves were nearly identical at water-jet angles of 30° and 60°; however, the 60° incidence conferred a slight advantage over 30° incidence. The section taper, section depth, and surface topography were optimized at a water-jet velocity of 24-m/s, 1.1-ms laser pulse width, 40-Hz frequency, and 180-A current. Under these conditions, the section taper and groove depth were 1.2° and 1.88 mm, respectively. The groove surfaces exhibited no splitting, slagging, or other defects, and no recast layers were visible.

**Keywords:** laser etching; water jet; polycrystalline silicon; orthogonal test

## 1. Introduction

Polysilicon material is a brittle semiconductor material with high wear resistance, hardness, chemical stability, and low thermal conductivity. The superior performance of silicon materials has secured their use in photovoltaics, aerospace, electronics, and other industries. However, owing to their physical properties, polysilicon and similar semiconductor materials are difficult to process effectively via conventional machining, i.e., the quality of processing may be poor. This problem can be resolved by employing laser processing technology. Such technology is widely used in electronic and other industries for precision machining of difficult-to-machine metal and non-metal material with high hardness, strength, toughness, and brittleness [1–3]. Unfortunately, under ambient air or inert gas conditions, the transient action of laser heat in the laser processing of materials inevitably produces microcracks and slag on the surface of the material, thereby reducing the qualification rate and processing quality of precision device products. To ameliorate the shortcomings of traditional lasers in processing such materials, liquid-assisted laser technology has been proposed herein. In this technology, with a hard and brittle material, the liquid reduces the temperature gradient at the processing site, thereby avoiding cracking and heat-affected areas of the material and improving the processing quality. Over time, liquid-assisted laser processing technology has branched into water-guided, underwater, and water-jet-assisted laser processing technology. Li et al. [4] etched trenches into silicon using

a water-guided laser micromachining method. They simulated the melting and removal of silicon under the cooling effect of the water jet and verified their model via comparative simulations and experiments. Their model considers the effects of cutting speed and other factors on the surface quality and heat-affected zone of the workpiece. Porter et al. [5] applied water-guided laser processing technology to cut metal sheets and summarized the influence of some processing parameters on the results. Hock et al. [6] cut stainless-steel plates and brass sheets (thickness ≤ 100 μm) using conventional laser and water-guided laser cutting technologies and compared the cutting widths, heat-affected zones of the two products. Compared to the conventional technique, the water-guided laser cutting system achieved a lower heat-affected zone, slag accumulation height, and slit width. Adelmann et al. [7] cut aluminum, titanium, steel, and other metals using a water-guided laser and investigated the effects of laser power, pulse repetition frequency and etching times on the processing quality. The maximum depths of the cut aluminum, titanium, and steel were 8, 4.7, and 1.5 mm, respectively, and the maximum aspect ratios were 66.7, 39.2 and 12.5, respectively. Using Nd:YAG pulse fiber lasers, Choubey et al. [8] conducted stainless-steel plate experiments in both air and underwater environments. Their results demonstrate that compared to air cutting, underwater cutting improves the sample and slightly reduces its slit width, heat-affected zone, and adhesion slag. Mullick et al. [9] established a model for investigating the energy-loss mechanism in underwater laser cutting. The model accounts for the interaction between the laser, material, and water. Most of the energy is lost by the scattering of the laser light in water vapor. The losses account for 40–50% of the total laser energy. The correctness of the model results was verified via experiments. Tsai [10] processed LCD glass materials using underwater and conventional laser techniques and investigated the effects of the laser parameters, material size, and liquid type on the removal rate, slit width, incision depth, taper, and surface roughness of the processed materials. Their results demonstrated the advantages of underwater laser processing over conventional laser processing. Charee et al. [11] processed silicon wafers via a flowing underwater laser composite etching process. To study the effects of underwater laser etching, they controlled the rate and direction of the water flow in a closed water chamber. With a high water flow rate, the grooves were deepened and few re-agglomeration layers were formed. The etching effect was superior to that obtained in still water. Kalyanasundaram [12] combined the parameters of laser water jets with those of hard and brittle processing materials and significantly improved the quality of the cut. Tangwarodomnukun et al. [13,14] compared the surface morphologies of samples prepared via hybrid laser–water jet micromachining, traditional silicon processing, and laser composite processing technologies. They varied the processing parameters and analyzed their effect on the heat-affected zone and processing quality. Bao et al. [15] established a fluid dynamics model based on smoothed particle hydrodynamics experimentally investigated the water-jet-assisted laser cutting of silicon materials. Experimental analysis revealed that the laser ablation of silicon mainly occurs via explosive melting. Zhu et al. [16] established a numerical model of heat transfer and material ablation in the water-jet-assisted laser etching of single-crystal germanium. They reported that the irradiated material can be discharged through the water jet and that during the non-pulsing periods, the water-jet cooling effectively removes heat buildup in the workpiece, thereby minimizing the thermal damage caused by laser heating. They also analyzed the influence of the processing parameters on the ablation process. Increasing the laser pulse energy deepened the grooves, whereas increasing the applied water pressure reduced the threshold workpiece temperature for material removal. Feng et al. [17] established a three-dimensional analytical model of the temperature field during the hybrid laser water-jet micromachining process. Their model considers the interaction between the laser, water jet, and workpiece. The absorption of laser light by the water, the formation of laser-induced plasma in water, the formation of bubbles, and the laser refraction at the air–water interface were discussed. To evaluate two machining methods, Hao Zhu et al. [18] compared the direct and chemical-assisted picosecond laser trepanning of single crystalline silicon. In their study, an orthogonal test design scheme was adopted to consider the relevant parameters affecting the trepanning process. They found

that the direct laser trepanning results are associated with significant thermal defects, whereas the chemical-assisted method can process microholes with negligible thermal damage.

Conventional gas-assisted laser processed materials have defects such as a large heat-affected zone and many re-casting layers. Compared with traditional gas-assisted laser processing, the laser processing area is cooled and impacted by the water jet. Specifically, during laser processing, the water jet cools the material and its impact can wash away the slag and recast a layerafter laser processing. The cooling action can reduce excess heat acting on the surface of the material, reduce the heat-affected zone, and reduce the generation of microcracks, thereby improving the surface quality of the material. In this paper, water jets with different angle and velocities were used to assist laser etching of polysilicon. Previously, we studied the effect of different water jet velocities on the processing grooves when the water jet angle was 30° [19]. However, to the best of our knowledge, the effects of changing both the water-jet angle and velocity on the depth and width of the etched grooves have not been investigated. This paper builds on previous studies by varying the jet angles and velocities in water-jet-assisted laser etching of polysilicon materials and examining its effect on the depth and width of the etched grooves. Subsequently, after fixing the water-jet angle at 60°, the optimal processing parameters in the experimental grooves are investigated in an orthogonal optimization experiment.

## 2. Theoretical Analysis of the Influence of Water-Jet-Assisted Laser Etching

Water-jet-assisted laser processing introduces a water jet to traditional laser processing. The auxiliary gas removes slag and debris generated in the laser processing, thereby improving the processing quality of the materials and reducing slag accumulation, the generation of microcracks, formation of recast layers during processing, and the heat-affected zone. Figure 1 schematizes the water-jet-assisted laser technique. Note that the water jet consumes part of the laser energy. In contrast, when jetted at appropriate velocity, the water jet washes away slag and the recast layer in an appropriate time, which reduces the heat-affected zone and improves processing quality. Water-jet-assisted laser processing also reduces photothermal efficiency and suppresses secondary adhesion of the etched material via laser-induced liquid cavitation, water cooling, and water flow.

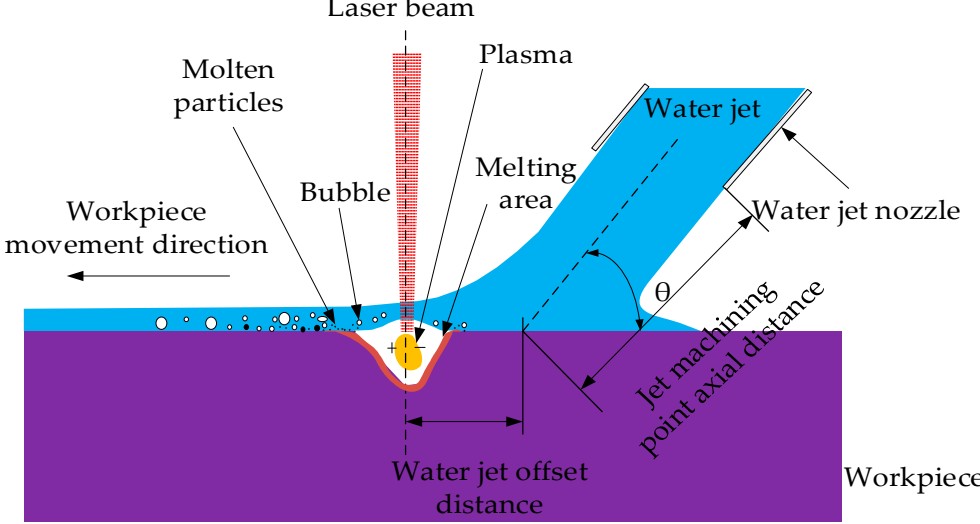

**Figure 1.** Schematic of water-jet-assisted laser processing.

Conventional gas-assisted and water-jet-assisted laser processing techniques remove material via laser irradiation, which smelts or vaporizes the material surface. In the conventional technique, most laser energy is absorbed by the material. The material's thermal properties will transfer excess heat around the laser action zone, which enlarges the heat-affected zone and increases slag accumulation. In water-jet-assisted laser etching technology, conventional laser etching is supplemented by water jets

applied at a certain angle and velocity. This technology combines the high efficiency of traditional gas-assisted laser processing with the advantages of water jets. The laser primarily acts as a softening material that melts or vaporizes the material as it is heated. The water jet strikes the laser processing area and cools it by heat transfer. Generally, the specific heat capacity of water is greater than that of the material matrix, which implies that water absorbs excess heat from the material surface and provides a cooling effect. In addition, increasing the water-jet velocity increases the impact force on the material. Within a certain range, the impact force is sufficient to flush the slag and other defects generated by the laser action. Then, the flowing water removes the slag or recast layer from the tank in sufficient time. Note that reducing slag and the recast layer from the bottom and walls of the trench can improve processing quality [20].

In this study, the effects of water jet pressure at different angles and velocities on the material were compared in fluent simulations. Herein, the water-jet angle was set to 30° or 60°, and the velocity was varied between 16, 20, 24, and 28 m/s (Figure 2). Note that many factors were simplified during simulation; thus, the results are theoretical rather than practical. As shown in Figure 2, increasing jet velocity at a fixed incident angle gradually increased the impact force of the water jet on the material. In addition, for a fixed water-jet velocity, the impact force was ~1.5 times higher at an impact angle of 60° than at 30°. Thus, it is theoretically demonstrated that 60° water-jet-assisted laser etching of the polysilicon material demonstrates groove depth and width greater than that of the 30° water-jet-assisted laser etching polysilicon material. In actual machining, greater jet velocity is advantageous for material removal; however, if the water flow velocity is excessive, the water jet will shake the experimental device, which will also deflect the impact center of the jet on the material. A significantly diverted water beam will cause splashing, which will generate "water mist" that affects the focused laser spot and incurs large laser energy loss.

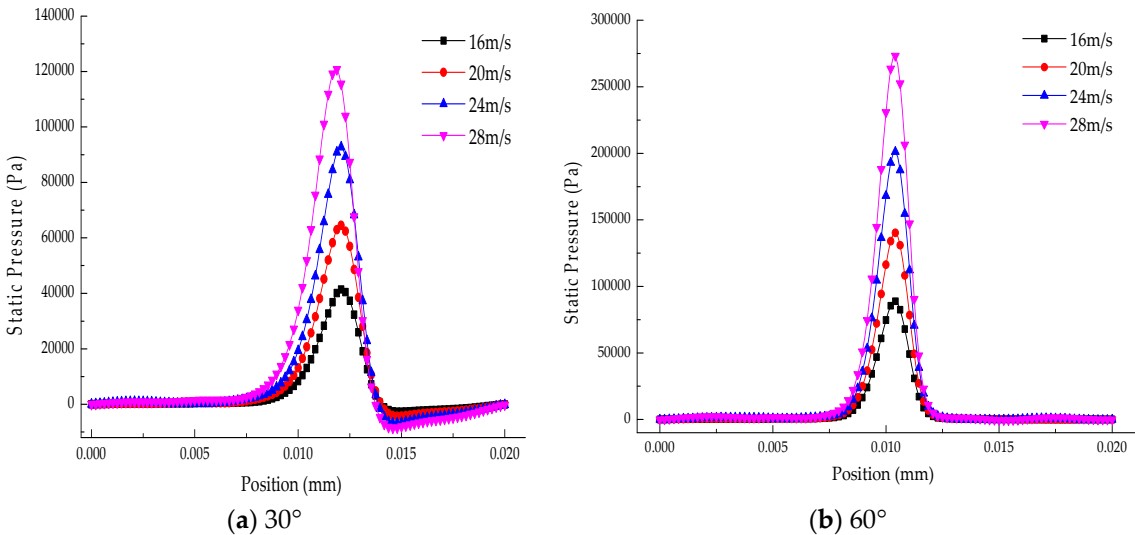

**Figure 2.** Impact force of a water jet striking a material at different velocities and impact angles.

## 3. Experimental Methods and Analysis

An HGL-LMY500 solid laser cutting and welding processing system (Wuhan Huagong Laser Engineering Co., Ltd., Wuhan, China) was used in the test. The HGL-LMY500 comprises an Nd:YAG solid-state pulsed laser (average output: 500 W), a power supply system, an optical system, a control system, a cooling system, a numerical control table, and a computer (Figure 3). During processing, this system produces a laser with a wavelength of 1064 nm, a multimode circular spot, and a spot diameter of 0.2 mm. By adjusting the current, pulse width, and repetition frequency, the laser output energy, laser light intensity, beam quality, etc. satisfy processing requirements. The main parameters of this laser system are shown in Table 1. Here, the water jet device is a water jet processing system designed

by our team. Note that the water-jet velocity and angle be adjusted. In addition, the water jet nozzle (water jet nozzle diameter is 0.7 mm) can be fixed to ensure that the water jet can impact and cool the material in the laser processing area. According to the actual work requirement, the motor, liquid tank, filter device, inlet pipe, plunger pump, overflow return control valve, pressure gauge, accumulator, one-way control valve, and jet nozzle and its fixing device.

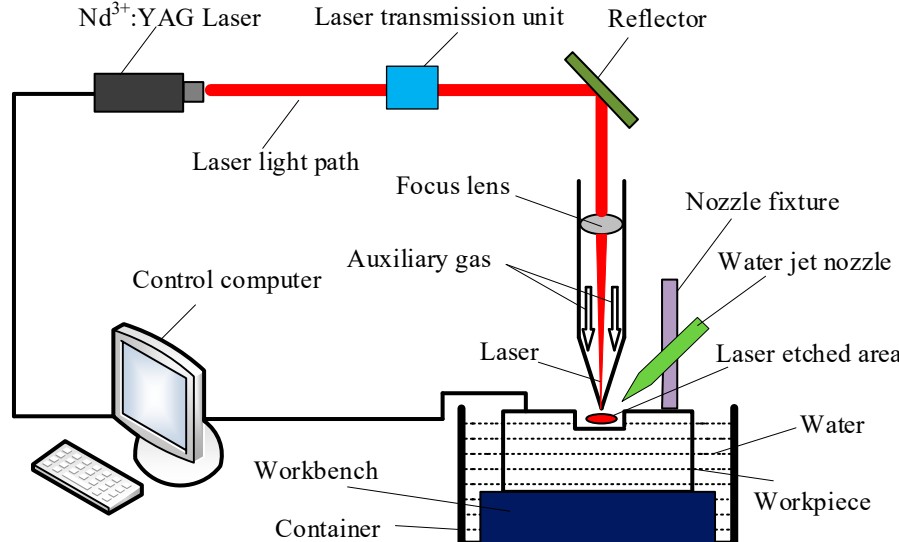

**Figure 3.** Schematic diagram of the experimental $Nd^{3+}$: YAG laser etching apparatus.

**Table 1.** Laser system processing parameters.

| Technical Parameters | Current (A) | Pulse Width (ms) | Repeat Frequency (Hz) | Single Pulse Energy (J) | Processing Speed (mm·s$^{-1}$) |
|---|---|---|---|---|---|
| Adjustment range | 100–400 | 0.1–20 | 0–150 | 0–90 | >0.1 |

The experimental sample was a polycrystalline silicon plate (length × width × height: $20 \times 20 \times 2$ mm$^3$). The effect of different water jet angles and velocities on the depth and widths of the processing grooves in water-jet-assisted laser etching was investigated. Prior to experiment, it is necessary to ensure that the following laser parameters remain unchanged: 40-Hz laser repetition frequency, 0.6-ms pulse width, 1-mm/s scanning speed, and 140-A current. Under this processing parameter, the laser irradiation intensity reaches $5.5 \times 10^5$ W/cm$^2$. When the irradiation intensity is $5.5 \times 10^5$ W/cm$^2$, it is sufficient to achieve the energy density required for irreversible damage to the surface material, and the material is removed by the action of the laser. In water-jet-assisted laser processing, the jet is impacted slightly behind the laser beam (1 mm behind processing area) to avoid direct contact between the jet and laser beam, avoid excessive loss of laser energy, and boost cooling and impact performance. The cross-sections and widths of the resulting grooves are shown in Figures 4 and 5, respectively. Figures 6 and 7 show the groove depths and widths under different water-jet velocity (horizontal axis) and impact angle (colors), respectively.

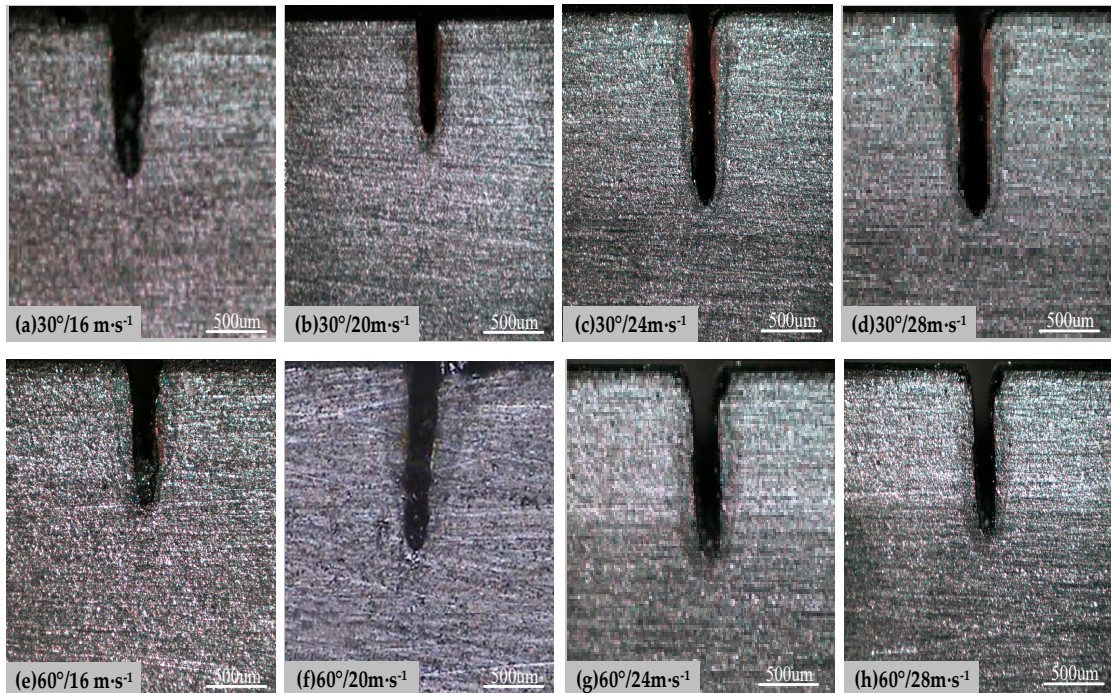

**Figure 4.** Cross-sections of grooves obtained by etching polycrystalline silicon at different impact angles (top row: 30°; bottom row: 60°) and jet velocities (left to right: 16, 20, 24, and 28 m/s) (25×).

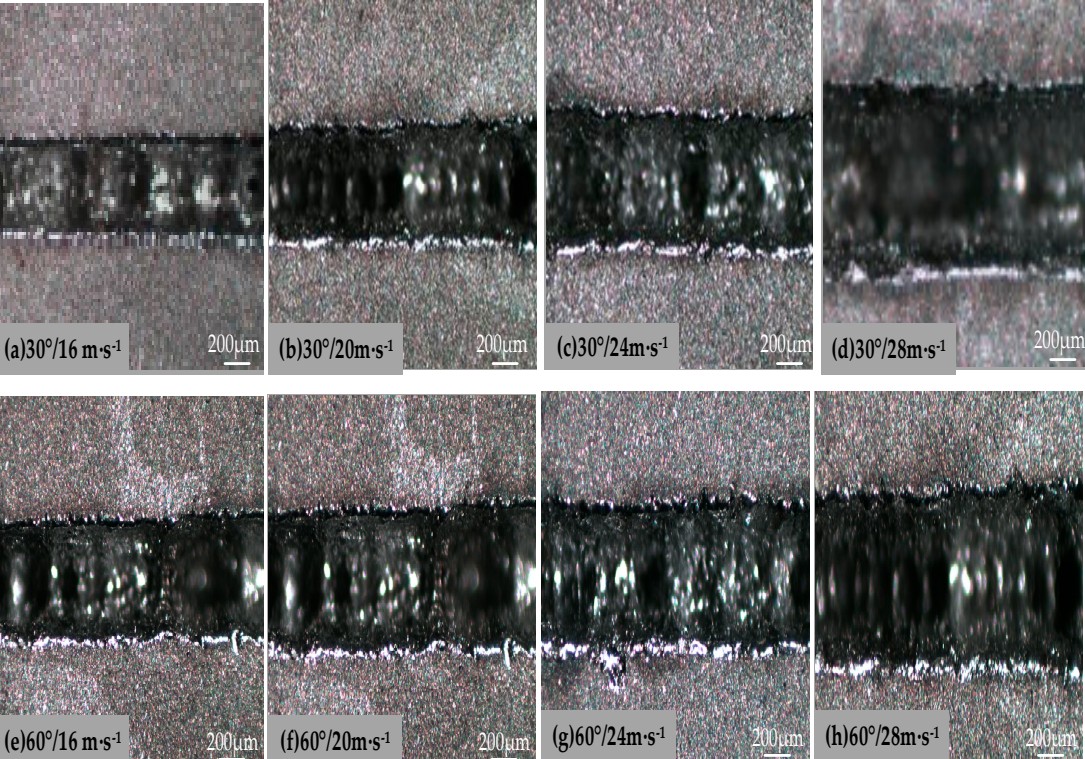

**Figure 5.** Widths of grooves obtained by laser etching at different impact angles (top row: 30°; bottom row: 60°) and jet velocities (left to right: 16, 20, 24, and 28 m/s) (30×).

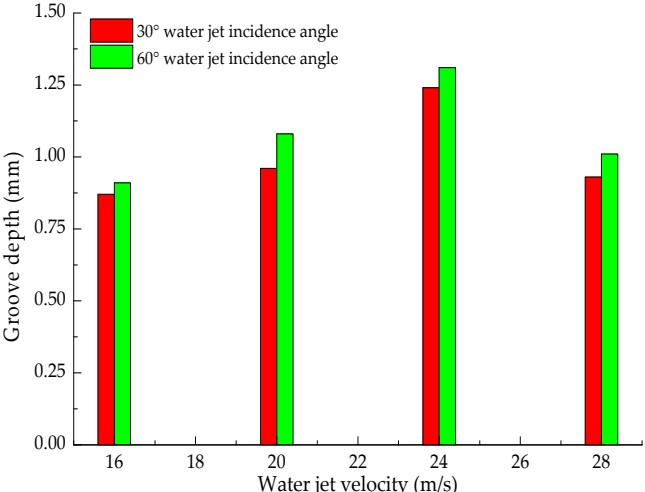

**Figure 6.** Groove depths obtained at different water-jet velocities (horizontal axis) and impact angle (colors).

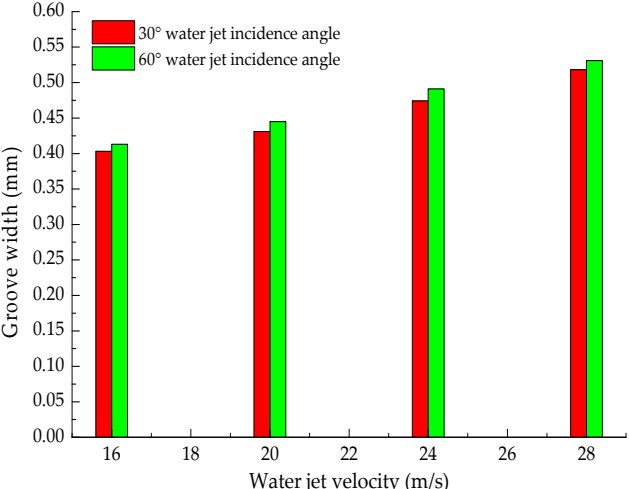

**Figure 7.** Groove widths obtained at different water-jet velocities (horizontal axis) and impact angles (colors).

As shown in Figures 4–7, the depth and width of the laser-etched grooves in the polysilicon material are dependent on water-jet velocity but not impact angle (the water-jet velocity trends were very similar at both 30° and 60°). The depth of the processing groove initially increased with increased jet velocity from 16 to 24 m/s but decreased between 24 and 28 m/s. In addition, the width of the processing groove gradually increased across the entire range of tested jet velocities. For the same water-jet velocity, the groove depths and widths were slightly larger at an impact angle of 60° than at 30°. The slight increase at 60° is attributable to the higher impact force at 60° than at 30°. As the water-jet velocity increased from 16 to 24 m/s, both the impact and cooling effect of the water-jet increased; however, the increasing trend of the cooling effect slowed down. As a result, the total increasing trend was dominated by the impact of the water jet, whereas the cooling effect played a secondary role. The molten silicon and slag deposited on the bottom of the tank were removed to reduce laser energy absorption; thus, the depth and width of the processing grooves increased gradually with water-jet velocity. However, at a water-jet velocity of 28 m/s, the large impact force was compromised because the experimental device was disturbed. At this velocity, the impact of the water jet deviated from the center of the jet impact, and the water beam diverged when striking the material surface, which caused splashes and "water mist" that affected the focused laser spot. In water-assisted laser processing, the material surface is covered with a thin, flowing layer of water, which absorbs some of the laser energy, thereby weakening the laser's action on the material. Simultaneously, the

enhanced convective heat transfer between the laser thermal energy and surrounding medium (i.e., the water) incurs large laser energy loss, which reduces the processing depth.

## 4. Orthogonal Test Design

### 4.1. Design Table of Orthogonal Test Plan

The orthogonal test method was employed to assess the quality of the water-jet-assisted laser etching of the polysilicon samples. In this experiment, the processing quality was primarily evaluated by examining the section taper, section morphology, and processing depth. At the largest processing depth, the processing tank must have the smallest taper and fine surface morphology. The cross-section of the sample trough was observed using a VM-3020E 2D imager, and the taper and depth of the section were measured. The results are shown in Figure 8.

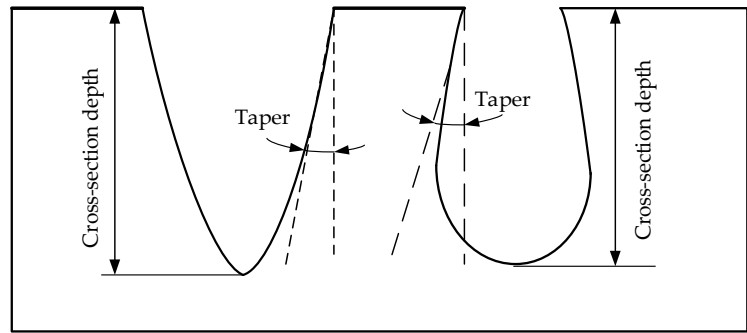

**Figure 8.** Cross-sectional profile of the groove in the polysilicon sample after processing.

Section 3 discussed the effect of water-jet angles and velocities on the depth and width of the grooves fabricated by laser-assisted etching of polysilicon material. Note that many interacting processing parameters are involved in the actual processing. The parameters that affect water-jet-assisted laser etching are the laser and water-jet parameters. To improve the etching effect, one or several sets of good combinations of processing parameters must be determined, which requires numerous experiments. The orthogonal test method optimizes the selection of processing parameters, which reduces the number of tests and the time required to select suitable processing parameters. As shown in Figure 2, the impact effect of the water jet at any incident velocity was slightly higher at 60° compared to that at 30°. Therefore, in the orthogonal test, the incident angle of the water jet was set to 60° and the other parameters were set according to those of the actual situation. Here, the main research objects were the laser processing parameters (laser pulse width, repetition frequency, and pulse energy) and pulse energy (represented by input current).

In the orthogonal test, the water-jet velocity, laser pulse width, laser repetition frequency, and laser input current were labeled *A*, *B*, *C*, and *D*, respectively, and each factor was assigned four levels (1, 2, 3, and 4), as shown in Table 2.

**Table 2.** Factor/level table.

| Level | Factor | | | |
|---|---|---|---|---|
| | *A* Water-Jet Velocity/m/s | *B* Pulse Width/ms | *C* Repeat Frequency/Hz | *D* Current/A |
| 1 | 16 | 0.5 | 30 | 150 |
| 2 | 20 | 0.7 | 35 | 160 |
| 3 | 24 | 0.9 | 40 | 170 |
| 4 | 28 | 1.1 | 45 | 180 |

The water-jet-assisted laser etching process was tested according to the L16 (45) orthogonal table (presented as Table 3). The design included 16 orthogonal experimental schedules. The subscripts in each plan indicate the level number (first column in Table 1). For example, A1 represents jet velocity of 16 m/s and $B_4$ indicates a pulse width of 1.1 ms.

**Table 3.** Orthogonal test table (L16 (45)).

| Test Number | *A* | *B* | *C* | *D* | Test Plan |
|:---:|:---:|:---:|:---:|:---:|:---:|
| 1 | 1 | 1 | 1 | 1 | $A_1B_1C_1D_1$ |
| 2 | 1 | 2 | 2 | 2 | $A_1B_2C_2D_2$ |
| 3 | 1 | 3 | 3 | 3 | $A_1B_3C_3D_3$ |
| 4 | 1 | 4 | 4 | 4 | $A_1B_4C_4D_4$ |
| 5 | 2 | 1 | 2 | 3 | $A_2B_1C_2D_3$ |
| 6 | 2 | 2 | 1 | 4 | $A_2B_2C_1D_4$ |
| 7 | 2 | 3 | 4 | 1 | $A_2B_3C_4D_1$ |
| 8 | 2 | 4 | 3 | 2 | $A_2B_4C_3D_2$ |
| 9 | 3 | 1 | 3 | 4 | $A_3B_1C_3D_4$ |
| 10 | 3 | 2 | 4 | 3 | $A_3B_2C_4D_3$ |
| 11 | 3 | 3 | 1 | 2 | $A_3B_3C_1D_2$ |
| 12 | 3 | 4 | 2 | 1 | $A_3B_4C_2D_1$ |
| 13 | 4 | 1 | 4 | 2 | $A_4B_1C_4D_2$ |
| 14 | 4 | 2 | 3 | 1 | $A_4B_2C_3D_1$ |
| 15 | 4 | 3 | 2 | 4 | $A_4B_3C_2D_4$ |
| 16 | 4 | 4 | 1 | 3 | $A_4B_4C_1D_3$ |

*4.2. Experimental Results and Optimization Options*

Following the test scheme shown in Table 3, an orthogonal test was conducted on the water-jet-assisted laser etching processing platform. Here, the measured data were the depth and taper of the groove of the given evaluation index. The 16 test results and their corresponding data are shown in Figures 9 and 10, respectively.

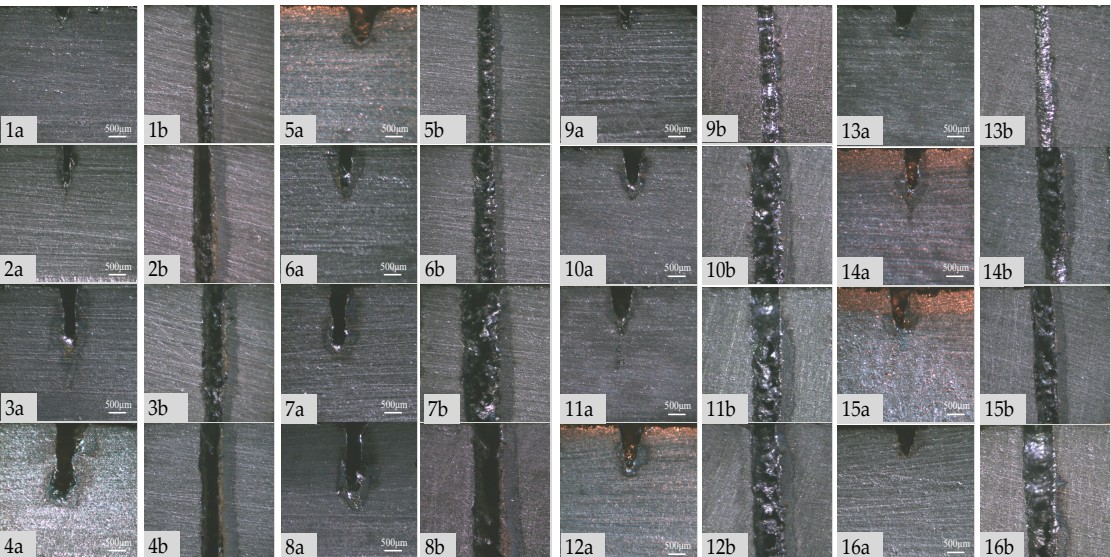

**Figure 9.** Orthogonal test results (15×). Label "a" represents the depth and taper of the section, and "b" represents surface topography.

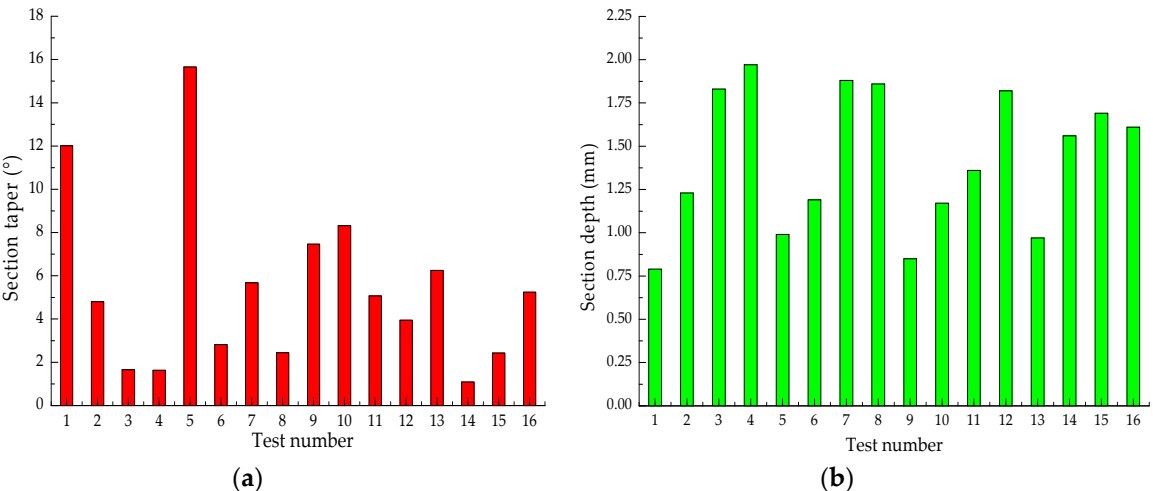

**Figure 10.** Experimental values of orthogonal tests 1–16. (**a**) Taper value; (**b**) Depth value.

The test results for each evaluation index in Table 4 were calculated and analyzed using the range analysis method. After computing test indices (i.e., section taper and section depth, corresponding to the *m*-th level of the *j*-th factor) and their average values ($K_{jm}$), the range differences $R_j$ of the two evaluation indexes were calculated for the *j*-th factor. The results are shown in Table 4.

**Table 4.** Range calculation values of orthogonal test results.

| Index | | A | B | C | D |
|---|---|---|---|---|---|
| Section taper/° | $\overline{K}_{j1}$ | 5.025 | 10.342 | 6.285 | 5.682 |
| | $\overline{K}_{j2}$ | 6.647 | 4.258 | 6.707 | 4.640 |
| | $\overline{K}_{j3}$ | 6.200 | 3.710 | 3.162 | 7.717 |
| | $\overline{K}_{j4}$ | 3.752 | 3.315 | 5.470 | 3.585 |
| | $R_j$ | 2.895 | 7.027 | 3.545 | 4.132 |
| Section depth/mm | $\overline{K}_{j1}$ | 1.455 | 0.900 | 1.238 | 1.513 |
| | $\overline{K}_{j2}$ | 1.480 | 1.288 | 1.433 | 1.355 |
| | $\overline{K}_{j3}$ | 1.300 | 1.690 | 1.525 | 1.400 |
| | $\overline{K}_{j4}$ | 1.458 | 1.815 | 1.497 | 1.425 |
| | $R_j$ | 0.180 | 0.915 | 0.287 | 0.158 |

From the average values of the indicators corresponding to each level of each factor, the appropriate level of each factor can be summarized as follows:

- If the required index must be as small as possible, we must consider the level corresponding to the smallest average.
- If the required index must be as large as possible, we must consider the level corresponding to the largest average.
- If the indicator must be moderate (a fixed value), we must consider the level corresponding to a moderate average.

To analyze the test results visually and intuitively and to obtain comprehensive conclusions, the relationships between the evaluation indicators and various factors (effect curves) were derived from the range analysis data. In this experiment, the average corresponding experimental indicator was plotted for each level in a given factor. The results of factors *A*, *B*, *C*, and *D* are shown in Figures 11–14, respectively.

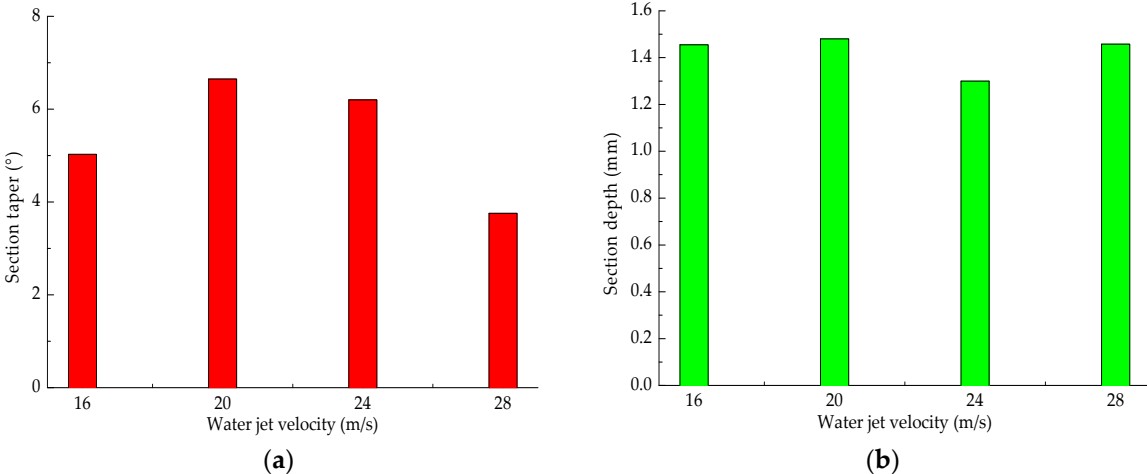

**Figure 11.** Effect of water-jet velocity on (**a**) cross-section taper and (**b**) cross-section depth.

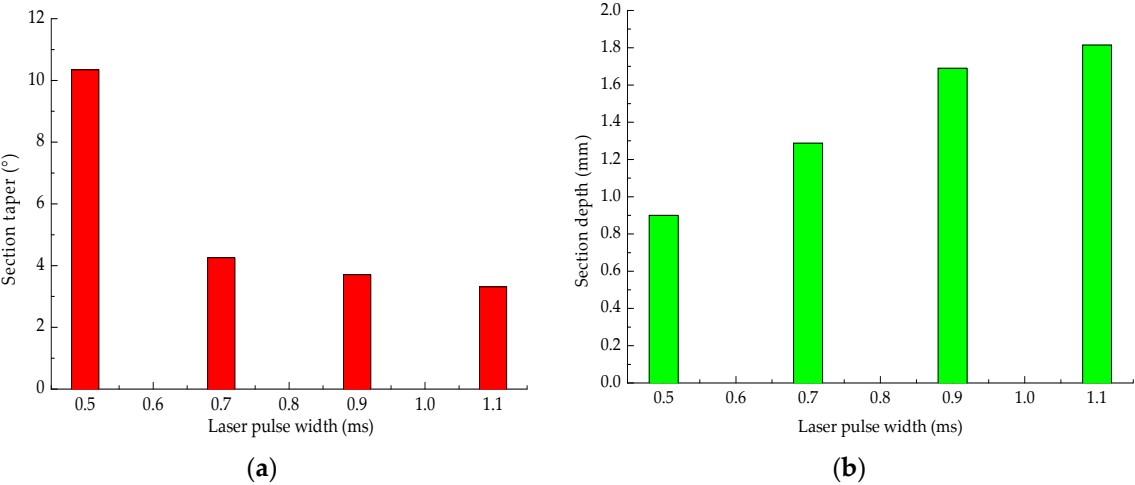

**Figure 12.** Effect of laser pulse width on (**a**) cross-section taper and (**b**) cross-section depth.

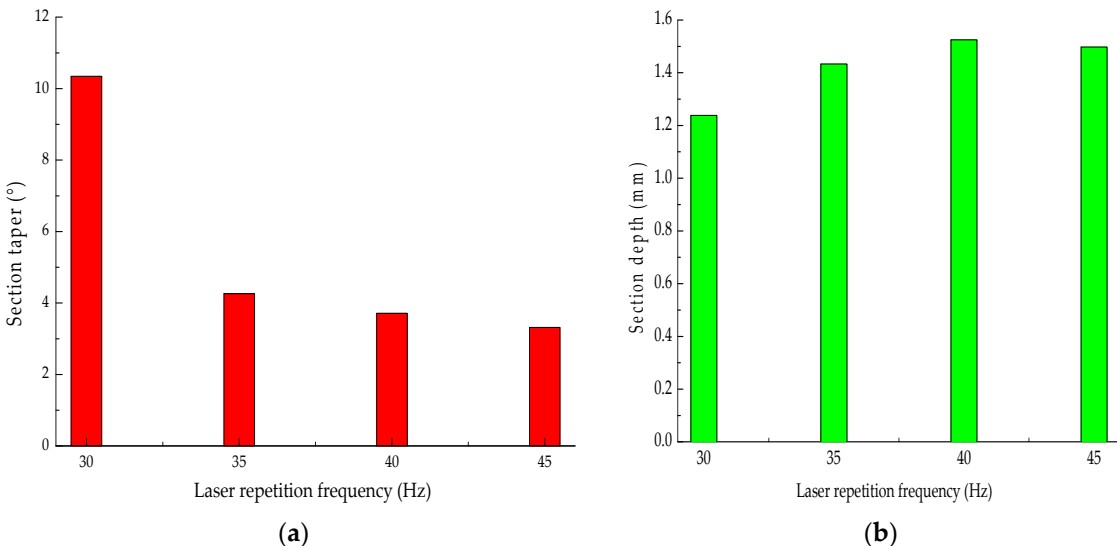

**Figure 13.** Effect of laser repetition frequency on (**a**) cross-section taper and (**b**) cross-section depth.

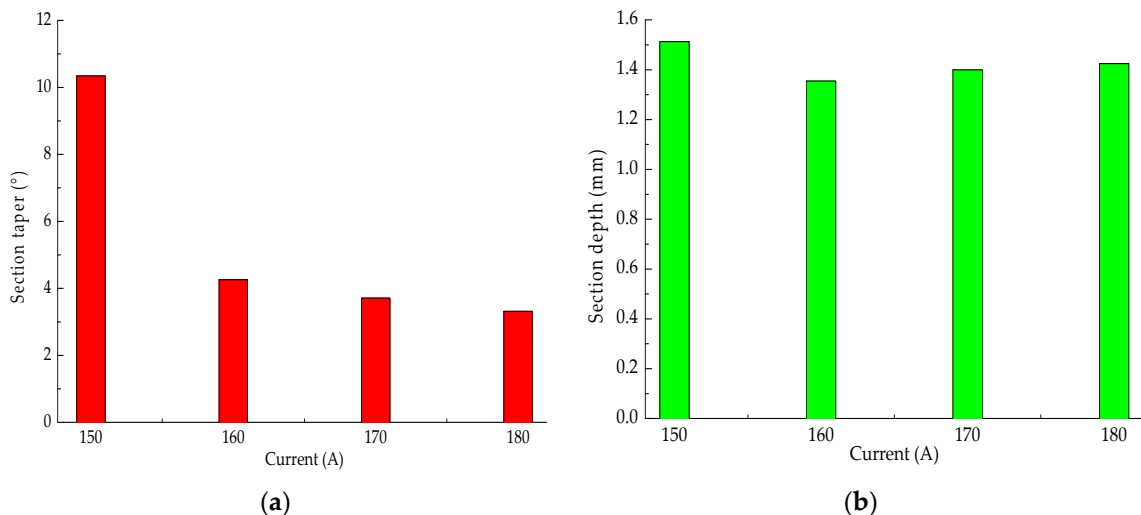

**Figure 14.** Effect of current on (**a**) cross-section taper and (**b**) cross-section depth.

The range analysis results given in Table 5 demonstrate that the section taper was most influenced by factor *B*, followed by factors *D*, *C*, and *A*. In addition, the actual processing requires that a smaller section taper will result in a larger section depth.

**Table 5.** Range analysis results of orthogonal test.

|  | Section Taper | Section Depth |
|---|---|---|
| Primary and secondary factors | *B, D, C, A* | *B, C, A, D* |
| Excellent level | $A_4, B_4, C_3, D_4$ | $A_2, B_4, C_3, D_1$ |
| Optimal combination | $B_4D_4C_3A_4$ | $B_4C_3A_2D_1$ |

The primary and secondary factors, optimal level, and optimal combination (i.e., the optimal solution) were obtained (Table 6) by combining the range calculation results given in Table 4 with the relationships between evaluation indicators and factors shown in Figures 11–14.

**Table 6.** Processing parameters in optimized schemes of each evaluation index.

| Evaluation Index | Actual Requirements | Excellent Solution | Water-Jet Velocity/m/s | Pulse Width/ms | Repeat Frequency/Hz | Current/A |
|---|---|---|---|---|---|---|
| Section taper/° | Smaller and better | $B_4D_4C_3A_4$ | 28 | 1.1 | 40 | 180 |
| Section depth/mm | Bigger and better | $B_4C_3A_2D_1$ | 20 | 1.1 | 40 | 150 |

As shown in Table 6, the optimal process that minimizes the cross-section taper combines a 1.1-ms laser pulse width, 40-Hz frequency, and 180-A current with the 28-m/s water-jet velocity. Under this condition, the cross-section taper after composite processing was 0.3° (Figure 15a). The optimum process that maximizes the tank depth combines a 1.1-ms laser pulse width, 40-Hz frequency, and 150-A input current with the 20-m/s water-jet velocity. With this configuration, the etched tank had a depth of 2.05 mm but was shaped like a teardrop with a non-ideal taper (Figure 15b).

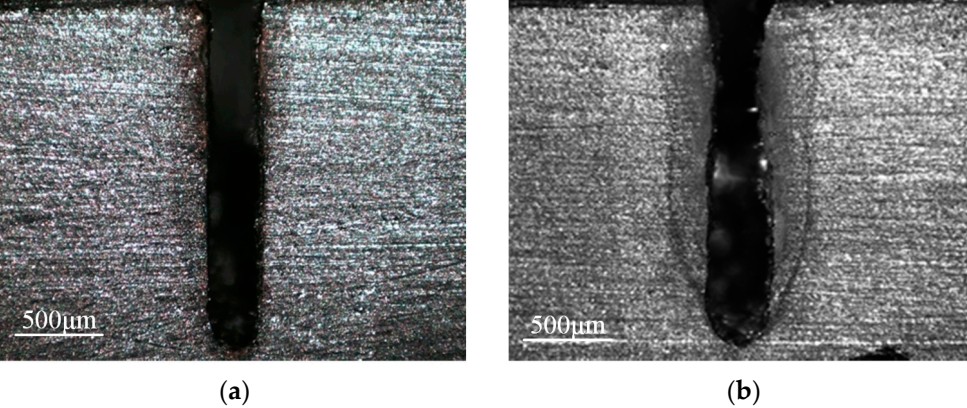

**Figure 15.** Cross-sectional morphology of samples after composite processing at processing parameters that optimize each evaluation index (×30): (**a**) section taper and (**b**) section depth.

Based on the above analysis, the final combination of optimal processing parameters was determined to be $B_4A_3C_3D_4$. The combination of processing parameters that optimized the cross-section taper and cross-section depth of the etched tank were as follows: jet velocity of 24 m/s, laser pulse width of 1.1 m/s, laser frequency of 40 Hz, and input current of 180 A. The cross-section and surface topography of the groove after composite processing under these conditions are shown in Figure 16. As can be seen, the surface of the groove body is free of splitting and slagging defects and has no recast layer. Here, the taper and depth of the groove were 1.2° and 1.88 mm, respectively. Although the etching result of each index was slightly worse than the result of optimizing a single indicator, it generally satisfies the etching effect requirements.

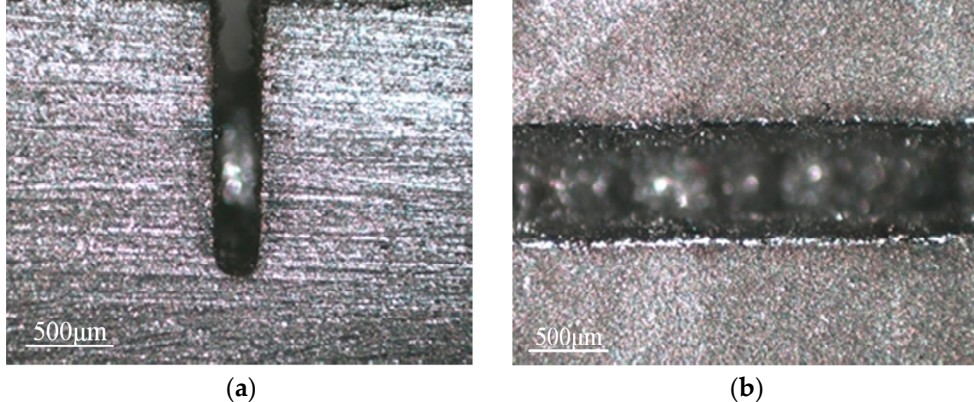

**Figure 16.** Surface and section morphologies of groove formed by composite machining under final optimized combination of processing parameters (×30): (**a**) sectional topography and (**b**) surface topography.

## 5. Conclusions

A water jet is frequently employed to assist laser etching of polysilicon materials. This study assessed the influence of water-jet incident angle and velocity on the depth and width of the etching groove. At a water jet incident at 60°, the effects of processing parameters (i.e., water-jet velocity, laser pulse width, repetition frequency, and input current) on the surface quality of the composite etching were investigated using the orthogonal test method. The processing parameters were optimized and verified. By comparing the verification test results with those of the orthogonal test, the processing parameters that optimize each evaluation index were obtained. Our primary findings are summarized as follows.

(1) The effects of 30° and 60° water-jet-assisted laser etching of polysilicon materials on the depth and width of the etched trenches were investigated. It was found that when the water jet angle was

maintained at one value, the depth of the machining tank initially increased and then decreased as water-jet velocity increased, and the width of the machining tank increased gradually. As the velocity increased from 16 to 24 m/s, the depth of the processed groove increased, but a further velocity increase (28 m/s) reduced the groove depth. The width of the processing groove was a gradually increasing function of the water-jet velocity. When the water-jet velocity was maintained at one value, the 60° water jet assisted the laser etching of the polysilicon material to obtain a tank with depth and width greater than that obtained by the 30° etching.

(2) Taking the 60° water-jet-assisted laser etching of polysilicon as an example, the orthogonal experimental method was used to optimize the processing parameters. In this study, the processing quality was evaluated relative to the depth, taper, and surface topography of the processing tank (the deeper the depth of the tank, the smaller the taper and the better the surface topography).

The experimental results demonstrated that the section taper was minimized under a 1.1-ms laser pulse width, 40-Hz frequency, and 180-A current when assisted by 28-m/s water-jet velocity. Here, the minimum section taper was 0.3°. The tank depth was maximized with a 1.1-ms laser pulse width, 40-Hz repetition frequency, and 150-A input current when assisted by a 20-m/s water-jet velocity. Here, the maximum tank depth was 2.05 mm; however, the section taper was not ideal. The optimal processing parameters were a laser pulse width of 1.1ms, repetition frequency of 40 Hz, water-jet velocity of 24 m/s, and input current of 180 A. Under these conditions, the cross-section taper and groove depth were 1.2° and 1.88 mm, respectively, and the groove surface demonstrated no defects.

**Author Contributions:** X.C. designed the study. Y.M., Y.Z., C.W., W.L. and L.H. developed the methodology. X.L. wrote manuscript.

**Funding:** This research was funded by Provincial Key Project of Natural Science Research Anhui Colleges (grant number: KJ2015A013 and KJ2015A050), Anhui Province Outstanding Young Talent Support Program Key Project (grant number: xyqZD2016153), National Natural Science Foundation of China (grant number: 51175229).

**Acknowledgments:** We thank all the authors for their joint efforts to complete the experiment. We also thank Jiangnan University for providing experimental equipment.

**Conflicts of Interest:** The authors declare no conflicts of interest.

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
