# Peer review of "Optimization of Processing Parameters for Water-Jet-Assisted Laser Etching of Polycrystalline Silicon"

_applsci, doi:10.3390/app9091882_

Round 1

Reviewer 1 Report

The manuscript discusses parametric studies of laser machining of polycrystalline silicon with water jet assistance. The main point is to determine a set or sets of parameters that enable consistent and reproducible results while focusing on the water jet angle and flow speed. Results are characterized in terms of machined trench width, depth, and taper angle. While laser machining with water assistance is a well-established technique, this work adds further parametric knowledge to the body of literature beyond mere incremental advancement.

The main body of the text is in very good shape overall, with only some minor issues which are specified below. However, the introduction concerning the context of current literature requires major revision. There is additional relevant work in this area that has been published but omitted. Please review the literature again. Waste less space on specific details that are irrelevant to the work at hand and redundantly stating the same conclusion—cleaner cuts and smaller heat-affected zones—multiple times. It would instead be more appropriate here to cite more of the published literature with a generalized overview of the experimental work that has already been carried out and the advantages/disadvantages of water-assisted laser machining observed. In the first paragraph, where the text reads, ‘transient action of laser heat inevitably generates microcracks and slag on the material surface,’ it should be noted that these effects result from processing under ambient air or inert gas conditions, as opposed to under liquid.

Specific issues with the citations: The phrases ‘chip thicknesses’ and ‘slit width’ are attributed to [5], but these terms never appear in that reference. The same reference is used three separate times: [6], [9], and [16], and many of the ‘theoretical equations’ cited as appearing in [6] do not appear there. In the cited values from [7], the aspect ratios appear in reverse order from the depths, and are thus not ‘respective’ as are the depths. The work cited as belonging to Behara and Sankar in [11] was performed by others; this reference is a review, not an original work. Delete the superfluous ‘is’ from before ‘30° [18].’ Prior to [19], the sentence reads, ‘Note that reducing slag and the recast layer from the bottom and walls of the tank can improve processing quality.’ However, there is no information on keeping tank walls and bottoms clear of debris in [19], or quality improvement gained therefrom. What effect do the tank walls and bottom have on the work piece, anyway? They are far away and not in contact. It appears this is a misstatement and should instead be ‘trench,’ not ‘tank’. The tank would be the container in Fig. 3; ‘trench’ the machined groove.

Page 3, near bottom: ‘As shown in Figure 3’ should read, ‘As shown in Figure 2.’ Why does Fig. 2 show ‘Static Pressure?’ The water is flowing, not simply resting on the work piece.

Page 4, first line: The text reads, ‘Thus, it is theoretically proven….’ Nothing has been ‘proven’ here. ‘Demonstrated’ perhaps, or ‘modeled.’ The end of the paragraph preceding Table 2 is a nonsensical fragment, and above this, the sentence needs a verb and should read, ‘Note that the water-jet velocity and angle may be adjusted.’

Page 5: The text states that the water jet impact point is situated 1 mm from the laser focus. How does this distance affect the processing? What if it is 2 mm, or 5 mm? This quantity also seems to be a valid parameter to vary.

Page 7, middle of top paragraph: The sentence beginning, ‘The slight increase at 60 °C…’ should be ‘The slight increase at 60° …’. At the end of paragraph the text reads, ‘which recues the processing depth.’ What does ‘recues’ mean? The authors seem to attribute the drop in depth at the highest flow speed to absorption of laser energy by the water, but isn’t it more likely that turbidity is disrupting the focus more and leading to loss of machining efficiency? The water is not absorbing significantly more laser energy than at lower flow speed; it would instead have to be much deeper to do that.

Remove hyphens and add spaces (consistently) between numerals and units. Ensure that all exponents are superscripted. Add spaces between text and brackets and punctuation and the next text letter throughout the paper and References.

Figures and tables: Figs. 1 and 9 may have issues with contrast visibility between black text and dark backgrounds. In Fig. 2, to what does the abscissa label ‘Position?’ refer? Position relevant to what? It is not clear. All figures with optical images need scale bars (Figs. 4, 5, 9, 15, 16). Table 1 and Fig. 3 have issues with column widths forcing line-wrapped letters, and in Table 6, force column heading ‘Actual require-ments’ to have the hyphen. Fig. 13 (a)’s ordinate label is lacking a unit. The Symbol Kji in Table 4 is not defined in the text, and the BCD values for the taper are all the same. Surely this is not correct, as the bar graphs in Figs. 11–14 show differing values.

Fig. 8 illustrates the difference between concave and convex tapers which implies a sign difference in the taper angle. However, Fig. 10 does not address a sign difference, and the text does not discuss whether both types of tapers were observed. If not, then clearly state only type of taper was observed. Otherwise, address the sign issue.

References: As previously noted, there are multiple issues with punctuation and spacing, and [6], [9], and [16] are all the same. The journal names/abbreviations contain errors in [1], [2], and [14]. [8] appears to be available only in Chinese (and “monocrystalline” is misspelled), so it will be of limited value to the general readership. If a translated version is available, please cite that or add a hyperlink to it. Author names are cited incorrectly in [10] and [11]. [19] is missing volume and article/page information. Include all authors’ names; only use ‘et al.’ for more than 10 authors. Please review the full MDPI Reference List and Citations Style Guide for consistent reference formatting.

Author Response

Reply The manuscript discusses parametric studies of laser machining of polycrystalline silicon with water jet assistance. The main point is to determine a set or sets of parameters that enable consistent and reproducible results while focusing on the water jet angle and flow speed. Results are characterized in terms of machined trench width, depth, and taper angle. While laser machining with water assistance is a well-established technique, this work adds further parametric knowledge to the body of literature beyond mere incremental advancement.

We would like to thank editor and reviewers for the comments and suggestions.After careful consideraions, we have revised the paper and prepared this detailed response.we hope to produce a stronger body of work based on the feeback. (Note: revisions are highlighted in red color in the manuscript)

Q:1.The main body of the text is in very good shape overall, with only some minor issues which are specified below. However, the introduction concerning the context of current literature requires major revision. There is additional relevant work in this area that has been published but omitted. Please review the literature again. Waste less space on specific details that are irrelevant to the work at hand and redundantly stating the same conclusion—cleaner cuts and smaller heat-affected zones—multiple times. It would instead be more appropriate here to cite more of the published literature with a generalized overview of the experimental work that has already been carried out and the advantages/disadvantages of water-assisted laser machining observed. In the first paragraph, where the text reads, ‘transient action of laser heat inevitably generates microcracks and slag on the material surface,’ it should be noted that these effects result from processing under ambient air or inert gas conditions, as opposed to under liquid.

Ans: Thank you for the comment. We have added relevant references in the introduction and made some modifications. Before the modification: transient action of laser heat inevitably generates microcracks and slag on the material surface Modified content: under ambient air or inert gas conditions, the transient action of laser heat in the laser processing of materials inevitably produces microcracks and slag on the surface of the material.

Q:2.Specific issues with the citations: The phrases ‘chip thicknesses’ and ‘slit width’ are attributed to [5], but these terms never appear in that reference. The same reference is used three separate times: [6], [9], and [16], and many of the ‘theoretical equations’ cited as appearing in [6] do not appear there. In the cited values from [7], the aspect ratios appear in reverse order from the depths, and are thus not ‘respective’ as are the depths. The work cited as belonging to Behara and Sankar in [11] was performed by others; this reference is a review, not an original work. Delete the superfluous ‘is’ from before ‘30° [18].’ Prior to [19], the sentence reads, ‘Note that reducing slag and the recast layer from the bottom and walls of the tank can improve processing quality.’ However, there is no information on keeping tank walls and bottoms clear of debris in [19], or quality improvement gained therefrom. What effect do the tank walls and bottom have on the work piece, anyway? They are far away and not in contact. It appears this is a misstatement and should instead be ‘trench,’ not ‘tank’. The tank would be the container in Fig. 3; ‘trench’ the machined groove.

Ans: Thank you for your suggestion. Before the modification:Hock et al. [5] cut stainless-steel plates and brass sheets (thickness ≤ 100 μm) using conventional laser and water-guided laser cutting technologies and compared the cutting widths, heat-affected zones, and chip thicknesses of the two products. Compared to the conventional technique, the water-guided laser cutting system achieved a lower heat-affected zone, slag accumulation height, and slit width. Modified content:Hock et al. [5] cut stainless-steel plates and brass sheets (thickness ≤ 100 μm) using conventional laser and water-guided laser cutting technologies and compared the cutting widths, heat-affected zones of the two products. Compared to the conventional technique, the water-guided laser cutting system achieved a lower heat-affected zone, slag accumulation height, and slit width. References 6, 9 and 16 and related content are deleted Before the modification:Adelmann et al. [7] cut aluminum, titanium, steel, and other metals using a water-guided laser and investigated the effects of laser power, pulse repetition frequency and etching times on the processing quality. The maximum depths of the cut aluminum, titanium, and steel were 8, 4.7, and 1.5 mm, respectively, and the maximum aspect ratios were 12.5, 39.2, and 66.7, respectively. Modified content:Adelmann et al. [7] cut aluminum, titanium, steel, and other metals using a water-guided laser and investigated the effects of laser power, pulse repetition frequency and etching times on the processing quality. The maximum depths of the cut aluminum, titanium, and steel were 8, 4.7, and 1.5 mm, respectively, and the maximum aspect ratios were 66.7,39.2 and 12.5, respectively. Before the modification:[11]Behera R,Sankar M R. State of the Art on Under Liquid Laser Beam Machining[J]. Materials Today Proceedings,2015,2(4-5):1731-1740. Modified content:Tsai C H , Li C C . Investigation of underwater laser drilling for brittle substrates[J]. Journal of Materials Processing Technology, 209(6):2838-2846. Deleted the original 18 of is. In the original , tank is changed to trench

Q:3.Page 3, near bottom: ‘As shown in Figure 3’ should read, ‘As shown in Figure 2.’ Why does Fig. 2 show ‘Static Pressure?’ The water is flowing, not simply resting on the work piece.

Ans: Thank you for your comment. ‘As shown in Figure 3’has been changed to ‘As shown in Figure 2.’ The static pressure in Figure 2 is mainly to study the effect of water jet impact materials on material removal rate at different angles and speeds.For example, the greater the water jet angle and speed, the greater the material removal rate.In this experiment, the depth and width of the groove are mainly considered, so the depth and width of the groove are larger.

Q:4.Page 4, first line: The text reads, ‘Thus, it is theoretically proven….’ Nothing has been ‘proven’ here. ‘Demonstrated’ perhaps, or ‘modeled.’ The end of the paragraph preceding Table 2 is a nonsensical fragment, and above this, the sentence needs a verb and should read, ‘Note that the water-jet velocity and angle may be adjusted.’

Ans: Thank you for your suggestion. ‘proven’has been changed to‘demonstrated’ In 4.1. Design table of orthogonal test plan is mainly the design of orthogonal experiments.Table 2 shows the four factors and the tables listed under the four levels, which are for Table 3.

Q:5.Page 5: The text states that the water jet impact point is situated 1 mm from the laser focus. How does this distance affect the processing? What if it is 2 mm, or 5 mm? This quantity also seems to be a valid parameter to vary.

Ans: Thank you for your suggestion. The jet action water jet slightly lags behind the laser beam at a position of 1 mm to act on the processing area, which not only provides good cooling and impact, but also avoids direct contact between the jet and the laser beam, resulting in excessive laser energy loss.Further research is needed for other locations.

Q:6.Page 7, middle of top paragraph: The sentence beginning, ‘The slight increase at 60 °C…’ should be ‘The slight increase at 60° …’. At the end of paragraph the text reads, ‘which recues the processing depth.’ What does ‘recues’ mean? The authors seem to attribute the drop in depth at the highest flow speed to absorption of laser energy by the water, but isn’t it more likely that turbidity is disrupting the focus more and leading to loss of machining efficiency? The water is not absorbing significantly more laser energy than at lower flow speed; it would instead have to be much deeper to do that.

Ans: Thank you for your suggestion. ‘The slight increase at 60 °C’has been changed to‘The slight increase at 60°’ ‘recues’has been changed to‘reduces’ ‘the drop in depth at the highest flow speed’,the explanation in the text is:‘However, at a water-jet velocity of 28 m/s, the large impact force was compromised because the experimental device was disturbed. At this velocity, the impact of the water jet deviated from the center of the jet impact, and the water beam diverged when striking the material surface, which caused splashes and “water mist” that affected the focused laser spot. In water-assisted laser processing, the material surface is covered with a thin, flowing layer of water, which absorbs some of the laser energy, thereby weakening the laser’s action on the material. Simultaneously, the enhanced convective heat transfer between the laser thermal energy and surrounding medium (i.e., the water) incurs large laser energy loss, which reduces the processing depth.Given your consideration‘it more likely that turbidity is disrupting the focus more and leading to loss of machining efficiency’,it is possible and needs further research.

Q:7.Remove hyphens and add spaces (consistently) between numerals and units. Ensure that all exponents are superscripted. Add spaces between text and brackets and punctuation and the next text letter throughout the paper and References.

Ans: Thank you for your suggestion. Related content has been modified in the text.

Q:8.Figures and tables: Figs. 1 and 9 may have issues with contrast visibility between black text and dark backgrounds. In Fig. 2, to what does the abscissa label ‘Position?’ refer? Position relevant to what? It is not clear. All figures with optical images need scale bars (Figs. 4, 5, 9, 15, 16). Table 1 and Fig. 3 have issues with column widths forcing line-wrapped letters, and in Table 6, force column heading ‘Actual require-ments’ to have the hyphen. Fig. 13 (a)’s ordinate label is lacking a unit. The Symbol Kji in Table 4 is not defined in the text, and the BCD values for the taper are all the same. Surely this is not correct, as the bar graphs in Figs. 11–14 show differing values.

Ans: Thank you for your comment. In Fig. 2, the position refers to the impact area of the water jet center.During the experiment, the central area of the water jet will slightly lag behind the laser processing area to ensure that the impact of the water jet is fully utilized. Figure 13 (a) has been modified. Table 4 has been modified. ‘After computing test indices (i.e., section taper and section depth) and their averages (corresponding to the m-th level of the j-th factor)’has been changed to‘After computing test indices (i.e., section taper and section depth,corresponding to the m-th level of the j-th factor) and their average values (Kjm)’ The relevant content of other parts has also been revised.

Q:9.Fig. 8 illustrates the difference between concave and convex tapers which implies a sign difference in the taper angle. However, Fig. 10 does not address a sign difference, and the text does not discuss whether both types of tapers were observed. If not, then clearly state only type of taper was observed. Otherwise, address the sign issue.

Ans: Thank you for your comment. The appearance of this shape is mainly due to the fact that the laser heat source model is a Gaussian heat source model. As shown in (a), the Gaussian heat source has a high temperature at the center and a low temperature on both sides. The appearance of this shape is partly due to the introduction of water jet.When the water jet angle is 60°, the impact force on the bottom of the tank body is larger when the water jet velocity is larger, so the bottom of the groove body is larger than the top of the groove body. On the one hand, this is to facilitate the reader to understand the method of taper measurement.The probability of the second shape appearing in actual processing is relatively small.On the other hand, the experimental numbers corresponding to the second shape appear are test1, 5, 9, 10, and 13 in the paper, respectively.

Q:10.References: As previously noted, there are multiple issues with punctuation and spacing, and [6], [9], and [16] are all the same. The journal names/abbreviations contain errors in [1], [2], and [14]. [8] appears to be available only in Chinese (and “monocrystalline” is misspelled), so it will be of limited value to the general readership. If a translated version is available, please cite that or add a hyperlink to it. Author names are cited incorrectly in [10] and [11]. [19] is missing volume and article/page information. Include all authors’ names; only use ‘et al.’ for more than 10 authors. Please review the full MDPI Reference List and Citations Style Guide for consistent reference formatting.

Ans: Thank you for your comment. The reference has been modified accordingly Reference 8 and its corresponding content have been deleted. Before the modification:Choubey A,Jain R K,All S, et a1.Studies on pulsed Nd:YAG laser cutting ofthick stainless steel in dry air and underwater environment for dismantling applications[J].Optics and LaserTechnology,2015(71):6-15. Modified content:Choubey, A. , Jain, R. K. , Ali, S. , et al. Studies on pulsed nd:yag laser cutting of thick stainless steel in dry air and underwater environment for dismantling applications. Optics & Laser Technology, 2015(71):6-15 Before the modification:Behera R,Sankar M R. State of the Art on Under Liquid Laser Beam Machining[J]. Materials Today Proceedings,2015,2(4-5):1731-1740. Modified content:Tsai C H , Li C C . Investigation of underwater laser drilling for brittle substrates[J]. Journal of Materials Processing Technology, 209(6):2838-2846. There are other content that has been modified and the text has been marked red.

Reviewer 2 Report

The paper entitled “Optimization of processing parameters for water-jet- assisted laser etching of polycrystalline silicon” reports the effects of changing both the water-jet angle and velocity on the depth and width of the etched grooves using an orthogonal test method. Although the effect of angle or velocity has been studied widely, the combination study is still novel and meaningful to the field. The results are presented logically, and discussions are convincing. However, there remain some questions, which need to be addressed before publication.

(1)  The previous work mentioned by the author is a little wordy. Instead of just listing their work one by one, a more logical and systematic summary may be better.

(2)  In Section 2, the benefit of water-jet-assisted laser processing was repeated too many times, i.e., reducing slag accumulation, recast layers and heat-affected zone, which makes this part long-winded.

(3)  In Section 3, “According to the actual work requirement, the motor, liquid tank, filter device, inlet pipe, plunger pump, overflow return control valve, pressure gauge, accumulator, one-way control valve, and jet nozzle and its fixing device.”. No predicate. Also, what is the role of the half-wave plate in the light path? Does polarization affect your results?

(4)  Scale bars are missing in all of the figures. For example, why the width of the grooves looks similar in Figure 4a and d, while they look pretty different in Figure 5a and d, although they are in the same magnification respectively.

(5)  In Page 7, the author claimed: “the depth and width of the laser-etched grooves in the polysilicon material are dependent on water-jet velocity but not impact angle (the water-jet velocity trends were very similar at both 30°and 60°).” The “not dependent on impact angle” should be revised since the author claimed the groove depths and widths were slightly larger at 60°and also the impact force is 1.5 times larger.

(6)  In Figure 8, there are two kinds of taper shown. Are the taper values both regarded as positive values in results? If considering the taper values for different shape of the taper as positive and negative, what the results in Figure 10a would be like?

(7)  In Page 8, “Here, the main research objects were the laser processing parameters (laser pulse width, repetition frequency, and pulse energy) and pulse energy (represented by input current).” What is the difference between these two “pulse energy”? Besides, the input current is not a straightforward and standard parameter for laser processing literature. Please convert it into pulse energy.

Author Response

The paper entitled “Optimization of processing parameters for water-jet- assisted laser etching of polycrystalline silicon” reports the effects of changing both the water-jet angle and velocity on the depth and width of the etched grooves using an orthogonal test method. Although the effect of angle or velocity has been studied widely, the combination study is still novel and meaningful to the field. The results are presented logically, and discussions are convincing. However, there remain some questions, which need to be addressed before publication.

We would like to thank editor and reviewers for the comments and suggestions.After careful consideraions, we have revised the paper and prepared this detailed response.we hope to produce a stronger body of work based on the feeback. (Note: revisions are highlighted in green color in the manuscript)

Q:(1)The previous work mentioned by the author is a little wordy. Instead of just listing their work one by one, a more logical and systematic summary may be better.

Ans: Thank you for your comment.According to your suggestion, the relevant content in the introductiont has been carefully revised.

Q:(2)In Section 2, the benefit of water-jet-assisted laser processing was repeated too many times, i.e., reducing slag accumulation, recast layers and heat-affected zone, which makes this part long-winded.

Ans: Thank you for your suggestion.

Before modificationThis technology combines the high efficiency of traditional gas-assisted laser processing with the advantages of water jets (high impact, transportation, and cooling effects), which reduces the area of the heat-affected zone, slag accumulation, and trough recast layers, and improves processing quality.

After modificationThis technology combines the high efficiency of traditional gas-assisted laser processing with the advantages of water jets.

Q:(3)In Section 3, “According to the actual work requirement, the motor, liquid tank, filter device, inlet pipe, plunger pump, overflow return control valve, pressure gauge, accumulator, one-way control valve, and jet nozzle and its fixing device.”. No predicate. Also, what is the role of the half-wave plate in the light path? Does polarization affect your results?

Ans: Thank you for your comment.

The laser processing system used is a commercial system in which the internal laser optics have been fixed and we cannot adjust it.Therefore, we did not consider the problem of half-wave plate polarization during the experiment.

Q:(4)Scale bars are missing in all of the figures. For example, why the width of the grooves looks similar in Figure 4a and d, while they look pretty different in Figure 5a and d, although they are in the same magnification respectively.

Ans: Thank you for your comment.

Scale bars have been added to the figure.

Figure 4a and d show the cross-sectional shape of the groove.The magnification is 25.When observing the cross-section, the sample is placed vertically for observation.

Figure 5a and d show the width of the slot and the magnification is 30.When observing the cross-section, the sample is placed horizontally and viewed from top to bottom.

Q:(5)In Page 7, the author claimed: “the depth and width of the laser-etched grooves in the polysilicon material are dependent on water-jet velocity but not impact angle (the water-jet velocity trends were very similar at both 30°and 60°).” The “not dependent on impact angle” should be revised since the author claimed the groove depths and widths were slightly larger at 60°and also the impact force is 1.5 times larger.

Ans: Thank you for your suggestion.

Before modificationAs shown in Figures 47, the depth and width of the laser-etched grooves in the polysilicon material are dependent on water-jet velocity but not impact angle (the water-jet velocity trends were very similar at both 30° and 60°).

After modificationAs shown in Figures 47, the depth and width of the laser-etched grooves in the polysilicon material are dependent on water-jet velocity and angle (the water-jet velocity trends were very similar at both 30° and 60°).

Q:(6)In Figure 8, there are two kinds of taper shown. Are the taper values both regarded as positive values in results? If considering the taper values for different shape of the taper as positive and negative, what the results in Figure 10a would be like?

Ans: Thank you for your comment. 

The appearance of this shape is mainly due to the fact that the laser heat source model is a Gaussian heat source model. As shown in (a), the Gaussian heat source has a high temperature at the center and a low temperature on both sides.

The appearance of this shape is partly due to the introduction of water jet.When the water jet angle is 60°, the impact force on the bottom of the tank body is larger when the water jet velocity is larger, so the bottom of the groove body is larger than the top of the groove body.

On the one hand, this is to facilitate the reader to understand the method of taper measurement.The probability of the second shape appearing in actual processing is relatively small.On the other hand, the experimental numbers corresponding to the second shape appear are test1, 5, 9, 10, and 13 in the paper, respectively.

Q:(7)In Page 8, “Here, the main research objects were the laser processing parameters (laser pulse width, repetition frequency, and pulse energy) and pulse energy (represented by input current).” What is the difference between these two “pulse energy”? Besides, the input current is not a straightforward and standard parameter for laser processing literature. Please convert it into pulse energy.

Ans: Thank you for your suggestion.

The laser processing system used is a commercial system in which the internal laser optics have been fixed and we cannot adjust it.The laser energy of a laser processing system can only be expressed by adjusting the magnitude of current, pulse width, and frequency.Therefore, only three variables are changed during the experiment to approximate the laser energy.
